# Parental Perception of Weight and Feeding Practices in SchoolChildren: A Cross-Sectional Study

**DOI:** 10.3390/ijerph18084014

**Published:** 2021-04-11

**Authors:** Patricia Inclán-López, Raquel Bartolomé-Gutiérrez, David Martínez-Castillo, Joseba Rabanales-Sotos, Isabel María Guisado-Requena, María Martínez-Andrés

**Affiliations:** 1Social and Health Care Research Center, Universidad de Castilla-La Mancha, 16071 Cuenca, Spain; patriciainlo@hotmail.com (P.I.-L.); jose_davidmcstllo@hotmail.com (D.M.-C.); maria.martinezandres@uclm.es (M.M.-A.); 2Faculty of Nursing, Universidad de Castilla-La Mancha, 02071 Albacete, Spain; joseba.rabanales@uclm.es (J.R.-S.); IsabelM.Guisado@uclm.es (I.M.G.-R.); 3Group of Preventive Activities in the University Field of Health Sciences, Universidad de Castilla-La Mancha, 02071 Albacete, Spain

**Keywords:** children, family, nutrition, weight perception, body image, obesity, feeding practices

## Abstract

Childhood obesity has become a public health problem. Parents play an important role in the transmission of feeding habits and the detection of their child′s weight status. The aim was to analyse the prevalence of overweight/obesity and to determine the relationship between children′s weight status, different feeding practices and weight misperception. A cross-sectional study was conducted in randomly selected schools. The children’s weight status was measured, and a questionnaire was used to identify the feeding practices applied by parents and their perception of their children′s weight. The sample comprised 127 children aged 4 and 5 years and 189 aged 10 and 11. Differences were observed between parental feeding practices and weight status, monitoring being the most used practice. Parents use less pressure to eat and more restriction if their children have overweight or obesity. Misperception of weight was 39.6%, being higher in overweight children, who were perceived as normal weight in 53.19%. Children classified as obese were perceived as overweight in 88.23%. The use of inappropriate eating practices shows a need for health education in parents according to weight status. In addition, the parents’ perception should be improved to increase early detection of overweight and start actions or seek professional help.

## 1. Introduction

Overweight and obesity are one of the main causes of disability and premature death at the global level. There are over 1.9 billion overweight/obese adults and over 380 million overweight/obese children in the world, with Europe being one of the most affected regions [1]. The probability of being obese in adulthood is greater in overweight or obese children, 77% of whom will become overweight and obese as adults [2]. Therefore, the prevention of obesity must begin during childhood, and is currently a public health priority [2,3,4,5,6].

Obesity is a complex problem [7] related to socioeconomic status, environments and infrastructure for play, access to healthy food and feeding practices, and closely linked to lifestyle [1,8,9,10,11,12]. Feeding practices and physical activity are established in childhood, and the family plays a central role in this process [4,6,13,14].

Parents are responsible for providing food, communicating eating habits and controlling the diet of their children [11]. It has been found that some of parental feeding practices on the eating behaviour of children may be counterproductive. Thus, pressuring children to eat and restricting the consumption of unhealthy foods has been associated with children’s inability to regulate their own food consumption [4,9,15]. This leads to an imbalance in the feeding practices of children and a potential increase in overweight, obesity and other eating disorders during their development [2,4,8,9,13,14,15].

Parents also play an essential role in controlling their children’s weight, through the proper detection of their children’s weight status according to clinical criteria and the start of appropriate actions [3]. However, the evidence indicates that parents’ recognition of overweight/obesity in children is limited. This judgment is not guided by objective measurements, but is instead influenced by various personal, social and cultural factors [2,16,17]. The most recent reviews suggest that parents’ underestimation of their children’s weight is a common phenomenon among parents of overweight/obesity children, despite the existence of societies with very different ideas of beauty [14,18]. The literature indicates an association between the incorrect perception of a child’s weight and less healthy diets, with the parents’ perception being an important indicator of children’s diets [4,14,16].

Context may affect the incorrect perception of parents, as they compare children through visual evaluation and are rarely guided by objective measures. Increasing levels of overweight in children are leading to increased weight categories being normalised. Therefore, the detection of overweight and obesity will be reduced in environments where overweight in children is high, which will impede parents from acting on this condition [3,6,7]. Given that there is proof of an association between the incorrect perception of children’s weight and less healthy diets, and that this incorrect perception depends on cultural values and the normalisation of excess weight, a better understanding of parents’ perception of the weight of children in our environment is required. Parents’ correct identification of their children’s overweight/obesity and knowledge of the factors that influence it may improve the way in which the risks of excess weight are communicated to all parents and increase commitment to behavioural interventions [2,18].

To the best of our knowledge, there is literature on trends in parents’ perception of their children’s weight based on nationwide surveys performed by the Spanish National Health Surveys [19,20,21]. Two of these three studies use self-reported weight and height, and all of them evaluated parental perception with this question: In relation to your child’s height, which of the following options best describes his/her weight: (1) substantially above normal, (2) slightly above normal, (3) normal, (4) below normal? There are two studies on how weight status affects parental feeding practices [4,22].

The aims of this study were (a) to analyse the prevalence of overweight and obesity among these children, (b) to ascertain the prevalence of parents’ incorrect perception of their children’s weight, and (c) to describe the relationship between the weight status of children of 4–5 and 10–11 years of age in the city of Albacete (Spain) and the parental feeding practices.

## 2. Materials and Methods

### 2.1. Study Design, Participants and Recruitment

A cross-sectional study was conducted with children aged 4, 5, 10 and 11 years enrolled at schools in Albacete, within a broader study for the validation of the questionnaire MapMe Tool, a body perception tool based on images to 4–5- and 10–11-year-old children, by Parkinson et al., in Newcastle, England [23].

To recruit participants, six public schools were selected by clustered sampling. The schools were situated in five different districts of Albacete, from diverse socioeconomic characteristics. The inclusion criteria were 4–5 and 10–11 years-old children, and children who could be measured. Two children could not be measured and weighted because of paraplegia and a sprain.

Sample size calculation was based on Comrey and Lee [24] to the validation and adaptation of questionnaires, the main aim of the cross-sectional study. They suggest that a good sample size in this type of study is 300. According to response rate in Spain, around 50% of participants do not respond to questionnaires [25]. A total of 622 children, 304 aged 4 to 5 and 318 aged 10 to 11, were invited to participate and 316 of those invited participated in the study.

### 2.2. Variables and Measurement Tools

#### 2.2.1. Anthropometric Variables

All measurements were made in standardised conditions, with two non-consecutive measurements by the same researcher.

-Weight: In kilograms, using a TANITA^®^ BC-418 MA (Tanita Corporation, Tokyo, Japan) bioimpedance scale.-Height: The height in centimetres was measured with a SECA^®^ 222 (SECA Corp., Hamburg, Germany) wall height rod.-Body mass index: Calculated as weight in kilograms divided by the square of the height in metres.

#### 2.2.2. Parental Perception

A single item was used where the father/mother was asked to indicate how they would describe the weight of their child: underweight, normal weight, overweight or very overweight (obese).

#### 2.2.3. Parental Feeding Practices

The Child Feeding Questionnaire (CFQ) [26] was administered, in its authorized Spanish version [4,27]. This scale is widely used to measure parental feeding practices. It is aimed at parents of children between 2 and 11 years of age. It comprises 7 subscales, 3 of which were used in the study. Those administered in the study focus on measuring parental control over child eating habits: restriction (limiting the amount and type of food), pressure to eat (being motivated or forced to eat) and monitoring (limiting unhealthy food consumption). It uses a Likert-type scale whose range is from 1 (never/ disagree) to 5 (always/agree). The score for each subscale is calculated with the mean score for the items.

#### 2.2.4. Educational Level

Both parents were asked about the highest educational level they had completed, classified as “primary education” if they belonged to one of these categories: (a) functionally illiterate, (b) without any studies or (c) had not completed primary education; as “middle education” if they had completed primary education, high school/secondary education or “*Bachillerato*” (two optional further years of high school, required if the student wishes to attend university or follow vocational education that prepares for work as a technician or in various fields); as “university education” if they had obtained an undergraduate, master’s or doctoral degree [28].

### 2.3. Procedure

After approval by the Clinical Research Ethics Committee, authorisation was requested from the Provincial Board of Education, Culture and Sport of Castille-La Mancha, and the management and School Boards of the selected schools.

The main researcher met with the director and teachers to explain the aims and methodology of the project. After accepting their participation, the questionnaire and the informed consent were given to the tutors, who distributed them to the children. They had to give it to their parents to fill out the questionnaire and sign the informed consent. Children returned the self-administered questionnaire and consent the measurement day.

The participants were recruited from January to April 2019. Data collection was conducted in school facilities from May to June 2019. The participants’ right to anonymity was respected.

### 2.4. Data Analysis

The IBM SPSS Statistics Version 24 program was used. The weight status variable was created in accordance with the criteria of the International Obesity Task Force (IOTF) [29]. Underweight students were included in the normal weight category, as this study examines overweight and obesity. To ascertain the prevalence of overweight and obesity, proportions were used. Chi-squared distribution was used to determine differences by sex, age, and parental level of education.

The CFQ factorial structure was checked in relation to the subscales included in this work. Given that the instrument had not been validated in Spain for children under 6 years old and that there is controversy in the literature on the magnitude of the restriction evaluated by the scale [26,27,30,31,32], it was decided to undertake an exploratory factor analysis. Analysis of main components was used with oblimin rotation, following Corsini et al. [31]. The results show a 4-factor structure, as two items that the authors include in the magnitude of restriction are grouped in a new factor, referring to the use of food as reinforcement by parents. We decided to exclude these two items from the scale of restriction, and they are not used in this work. The reliability of the scale is α = 0.81; restriction α = 0.87; pressure to eat α = 0.79; monitoring α = 0.85. None of the scales shows a normal distribution, and therefore, it was decided to use non-parametric tests.

## 3. Results

### 3.1. Sample Characteristics

Table 1 shows the descriptions of the sample and the weight status. The final sample was made up of 127 children aged 4 and 5 years (40.2%) and 189 aged 10 and 11(59.8%); 52.8% were boys. The non-response or rejection rate was 58.2% for the 4- and 5-year-olds and 40.5% for the 10- and 11-year-olds. Non-response was especially high in one of the schools, located in an area described by the institutions as of low socioeconomic status.

### 3.2. Weight Status

The prevalence of overweight and obesity was much higher among children of 10 and 11 years old (χ^2^ = 30.147, *p* < 0.00) (Table 2).

As shown in Table 3, a high level of education is related to greater prevalence of low or normal weight, both in fathers (χ^2^ = 12.858, *p* = 0.012) and mothers (χ^2^ = 22.153, *p* = 0.000).

### 3.3. Parental Perception of Weight Status

Table 4 shows the parents’ perception by sex, age and weight status of the children. There are no significant differences by sex (χ^2^ = 4.651, *p* = 0.098), but there are by age (χ^2^ = 74.649, *p* = 0.000) and weight status (χ^2^ = 184.371, *p* = 0.000).

Children classified as overweight were perceived as normal weight (53.19%). Children classified as obese were perceived as overweight (88.23%). With regards to extreme incorrect perception, only 1 out of 17 cases of children with obesity (5.89%) were classified as normal weight.

### 3.4. Parental Feeding Practices

We were interested in ascertaining whether there existed a relationship between parents’ correct and incorrect perceptions of the weight status of children with overweight and obesity and eating habits. The Mann–Whitney U test revealed no relationship between the perception or any of the eating habits studied.

### 3.5. Correlations

Finally, Spearman’s correlations were used to evaluate relationships between weight status, parents’ educational level and each CFQ subscale (Table 5). There was a high correlation between restriction and monitoring, which is also associated with increased weight. Pressure to eat was negatively related to the child’s weight status.

## 4. Discussion

Childhood obesity is a public health issue that is difficult to address due to multiple influencing factors and the lack of knowledge of the processes that lead to the action or inaction of parents. This study was intended to describe parental feeding practices in schools of the city of Albacete based on their weight status, as well as parents’ incorrect perception of their children’s weight.

The prevalence of overweight and obesity in our sample is similar to national data [19,33,34,35,36,37]. However, in the 4-to-5-year-old group, prevalence is lower in our study compared with others, including one conducted in Castilla-La Mancha, the autonomous community in which Albacete is located [36]. The high rate of non-response among 4–5-year-old pupils might have affected these results. It may also be that greater awareness of weight is affecting levels of overweight in younger children [36], or the non-response could be related to the fact that the schools with less participation were those which the socioeconomic level was low [38]. Conversely, the significantly greater prevalence of overweight and obesity among 10–11-year-old pupils matches the epidemiological data, indicating that overweight increases from 6 years of age [19,34,35]. Comparing the weight status by sex, normal weight and overweight percentages are similar among boys and girls, but there is greater prevalence of obesity between boys. There is no consensus in the literature about the relationship between weight status and sex [33,34]; this might be explained by the influence of cultural issues. In this sense, authors show that greater prevalence of obesity in boys may be related to greater social acceptance of large bodies in males than in females. 

In terms of parental level of education, the lower this education level, the greater is the prevalence of overweight and obesity in the children, which is consistent with the literature [2,8]. In a study carried out by Almoosawi et al. [2], an association was also observed between the maternal education level and healthy feeding practices. This reveals the need to improve understanding of nutrition and health, particularly in the population of a lower socio-educational level, and to appropriately adapt strategies.

Our primary interest was to determinate parents’ perception of their children’s weight. In the literature, misperception varies from one study to another, from 12–18% [18] up to 48% [14]. In our case, it reaches a considerably high 40%, which may be affected by cultural aspects. In the case of children with overweight and obesity, the underestimation of weight is 63%, which may correlate with the normalisation of overweight in our environment. In any event, there is a common trend of parents underestimating the weight of their children when they are overweight or obese, while children of normal weight are usually perceived correctly [2,14,17]. However, the underestimation is not extreme, and thus, children with obesity are perceived as overweight, which indicates that parents identify their weight to be inappropriate, but not seriously so.

The identification of an incorrect perception of weight is important, as it has been found that children whose weights are perceived correctly by their parents have healthier diets than those perceived incorrectly [2]. Additionally, the correct perception of children’s weight is associated with a lower increase in weight over time [2]. However, a recent study has found that the detection of overweight in children aged 4 years by parents did not protect them against gaining weight at 13 years of age [14]. This indicates that correct perception, without other interventions, does not prevent overweight or obesity.

Several studies have found similar levels of underestimation in boys and girls [18,39]. In line with our findings, studies that found differences showed a parental tendency to underestimate the weight of their sons. This could relate to the differences in body composition, but could also indicate the influences of social norms. These press women to have a small body and increase the probability to of earlier identification of overweight in females [39], while males having bigger bodies might be seen as a physical advantage [14].

Despite some studies reporting that the level of education of parents has little impact on their perception of their children’s weight, other studies show that parents with a low educational level have a higher rate of incorrect perception [2,39]. Our study also shows this association, having found a greater correct perception among parents with a higher level of education. Again, this suggests that, in educating parents, greater effort must be made with those of a lower level of education. Thus, parents’ education is fundamental to the perception of overweight/obesity. There is no consensus about what types of educational strategies could improve the perception of weight status in children [2,39].

Furthermore, we were interested in ascertaining the parental feeding practices and their relationship with weight status. Our results show that the use of restriction is greater among children with overweight and obesity. The literature indicates that restrictive practices influence overweight/obesity because they impede the self-regulation of eating among children, thereby increasing the risk of gaining weight in the future or eating when not hungry. For example, restricting “junk food” increases subsequent consumption when parental control is lifted, causing an increase in weight [26,40]. Other studies indicate that parents tend to focus only on restricting one aspect of diet, such as consuming less sugar, instead of promoting healthy feeding practise [2].

It has been suggested that the use of restriction facilitates the underestimation of overweight and obesity by parents [4,18,26,27,31,32], with the belief that they are thereby adequately controlling their eating, which leads to underestimating the real weight of their children [4,31,40]. In our work, the use of restriction is associated with overweight and obesity among children, but not with a greater incorrect perception of weight. In this sense, it seems that there would be a greater effect on the ability to regulate children than on parents’ perception.

Regarding the pressure to eat, concern for children being underweight leads parents to pressure children to increase consumption [4,26,27,31,32], and this seems to occur in our data. However, the subscale is oriented toward increasing the consumption of food in general, not promoting healthy feeding; therefore, it has been detected that the pressure to eat healthy foods reduces children’s preference for them [26,40,41]. Therefore, this also appears not to be a recommended practice, although this could not be verified in our work. Thus, it would be necessary to use a tool that allows to measure children’s eating preferences and their relationship with parent feeding practices.

Finally, monitoring, which is the most commonly used practice, barely has an impact on the handling of overweight or obesity of children. This would indicate that mere supervision does not lead to significant behavioural changes in terms of children’s feeding practices and that other more active practices would be preferable in terms of family feeding practices [4,11,12,26,27,31,32].

### 4.1. Limitations

This research has a series of limitations. Being a cross-sectional study, the direction of the relationships could not be determined, and therefore, we must be cautious with the results. Additionally, a large prevalence of parents with higher studies was found, due to the high rate of non-response among families of a low socio-economic level, which may have biased the findings along with the sample size. Despite this, among the strengths of this study, it must be highlighted that this is the first study to address parents’ perception of their children’s weight in Spain related to parental feeding practice strategies.

### 4.2. Implications for Research and Practice

Increasing levels of overweight in children are leading to increased weight categories being normalised. Therefore, the detection of overweight and obesity will be reduced in environments where overweight in children is high, which will impede parents from acting on this condition. Health professionals and especially nurses have a central role in the promotion of healthy habits and prevention of chronic illness related to overweight and obesity.

Thus, a better understanding of parents’ perception of the weight of children in our environment is needed. The use of inappropriate strategies and the underestimation by parents in our sample is similar to that found in other studies and alerts us to the need to work on parents’ education on health. This should be focused on healthy eating habits rather than on generating concern about weight, particularly with parents with a low level of education. Additionally, and along the same lines, in Primary Health Care, monitoring consultations should systematically ask about parents’ perception of weight in an attempt to review it and increase early detection.

## 5. Conclusions

This work shows that the relationship between overweight/obesity in children and the practices and perceptions of parents is complex. A better understanding of this association is necessary to implement effective preventive actions. Parents use eating habits with their children that may be counterproductive (as is the case with restriction) or which have no effect (such as monitoring). Although this work could not establish a relationship between concern for weight and the practices used, existing studies have demonstrated that the more concerned parents are about overweight/obesity, the more they use these practices. The belief that they are doing something to improve their children’s eating habits and the incorrect perception of their children’s weight would explain their not attempting to correct such habits or asking for help, despite their efforts not having the expected effects on weight. More studies would be necessary to test these beliefs in Spain in the future, as social and cultural issues affect these relationships. Undertaking longitudinal studies to study causality would also be an interesting line of research with a representative sample size and ensuring a representation of families with all socioeconomic status.

## Figures and Tables

**Table 1 ijerph-18-04014-t001:** Sample characteristics.

			Total	Male	Female	*p*
Age (years)	4		20.3%	20.4%	20.1%	0.37
	5		19.9%	22.8%	16.8%	
	10		32.9%	33.5%	32.2%	
	11		26.9%	23.4%	30.9%	
Weight	4–5 years	Mean	19.0051	19.0632	18.9291	0.85
		SD	(4.01)	(3.65)	(4.47)	
	10–11 years	Mean	41.0296	41.4042	40.6511	0.75
		SD	(9.81)	(10.32)	(9.31)	
BMI	4–5 years	Mean	14.3535	14.2720	14.4602	0.59
		SD	(1.89)	(1.65)	(2.18)	
	10–11 years	Mean	19.0463	19.2915	18.7985	0.49
		SD	(3.55)	(3.88)	(3.20)	
Weight status (IOTF)	Underweight/normal weight		79.4%	80.2%	78.5%	0.69
	Overweight		15.2%	13.8%	16.8%	
	Obesity		5.4%	6.0%	4.7%	
Educational level	Father	Primary	36.6%	37.7%	35.3%	0.87
		Middle	28.3%	27.2%	29.5%	
		University	35.2%	35.1%	35.3%	
	Mother	Primary	22.2%	20.2%	24.3%	0.49
		Middle	29.9%	32.5%	27.0%	
		University	47.9%	47.2%	48.6%	

Abbreviations: SD (standard deviation). IOTF (International Obesity Task Force).

**Table 2 ijerph-18-04014-t002:** Weight status by age and sex.

	Total	Age					
		4–5 Years			10–11 Years		
		Total (4–5 Years)	Male	Female	Total (10–11 Years)	Male	Female
Underweight/normal weight	79.4%	93.7%	94.5%	92.7%	69.8%	68.7%	70.2%
Overweight	15.2%	3.1%	4.1%	1.8%	23.3%	21.9%	25.5%
Obesity	5.4%	3.1%	1.4%	5.5%	6.9%	9.4%	4.3%

**Table 3 ijerph-18-04014-t003:** Weight status by parental educational level.

	Father Educational Level				Mother Educational Level			
	Primary	Middle	University	*p*	Primary	Middle	University	*p*
Underweight/normal weight	33.2%	28.4%	38.4%	0.012	19.0%	27.4%	53.6%	0.000
Overweight	42.2%	33.3%	24.4%	0.012	29.8%	46.8%	23.4%	0.000
Obesity	76.9%	7.7%	15.4%	0.012	50.0%	18.8%	31.3%	0.000

**Table 4 ijerph-18-04014-t004:** Parental perception by sex, age and weight status according to International Obesity Task Force (IOTF) criteria.

		Parental Perception		
		Correct	Overestimate	Underestimate
**Sex**	Male	57.2%	30.1%	12.7%
	Female	63.9%	19.7%	16.3%
**Age**	4–5 years	44.1%	50.4%	5.5%
	10–11 years	71.5%	8.1%	20.4%
**Weight status**	Underweight/Normal weight	66.7%	31.7%	1.6%
	Overweight	46.8%	0.0%	53.2%
	Obesity	5.9%	0.0%	94.1%
**Total**		60.4%	25.2%	14.4%

**Table 5 ijerph-18-04014-t005:** Correlations between weight status, parental educational level and parental feeding practices, controlling for the effect of sex and age variables.

	1	2	3	4	5	6
Weight status	-	−0.130 *	−0.145 *	0.231 **	−0.272 **	0.125 *
2. Father educational level		-	0.556 **	−0.068	−0.183 **	0.079
3. Mother educational level			-	−0.064	−0.218 **	0.024
4. Restriction				-	0.176 **	0.296 **
5. Pressure to eat					-	0.035
6. Monitoring						-

* *p* ≤ 0.05, ** *p* ≤ 0.01. Spearman rank correlation.

## Data Availability

Data available on request due to restrictions privacy and ethical. The data presented in this study are available on request from the corresponding author. The data are not publicly available due to ethical restriction following recommendation of Clinical Research Ethics Committee.

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
