# Peer review of "Parental Perception of Weight and Feeding Practices in SchoolChildren: A Cross-Sectional Study"

_ijerph, 2021, doi:10.3390/ijerph18084014_

Round 1

Author Response

REVIEWER 1

Authors: Thank you very much for your comments. We honestly think that these comments have greatly improved the manuscript.

1.- Grammatical errors

Response 1: Following the reviewer’s suggestion, we have changed all the errors:

Line 12

Childhood obesity has become a public health problem.

Line 18-19

The sample was comprised of 127 children aged 4-5 years and 189 aged 10-11.

Lines 18-19

The sample was comprised of 127 children aged 4 and 5 years and 189 aged 10 and 11.

Lines 29-30

Overweight and obesity are one of the main causes of disability and premature death at the global level.

Line 164

Table 1 shows the descriptions of the sample and the weight status.

2.- This sentence is in first person. Please use third person throughout.

Response 2: Following the reviewer’s suggestion, we have changed the sentence:

Lines 95-100

To calculate the sample size, It was based on Comrey and Lee[24] to the validation and adaptation of questionnaires, the main aim of the cross-sectional study. They suggest that a good sample size in this type of study is 300. According to response rate in Spain, around 50% of participants do not respond to questionnaires[25]. A total of 622 children was invited to participate, 304 pupils aged 4 to 5 and 318 aged 10 to11.

Reviewer 2 Report

General comments: Thank you for the opportunity to review this manuscript. I find this study very interesting and I congratulate you for your work. It focuses on an important topic, relevant to public health and children health. Please see my comments below.

Introduction section:

  • Page 1 line 30: According to the first reference cited, WHO reports an increase of overweight and obesity worldwide except in regions of Africa and Asia. So, I suggest to modify this sentence: “…Europe being the most affected region”.

  • Page 1 line 40: Remove the question mark after “eating behaviour of children

Materials and Methods section:

  • Page 2 lines 89-90: Could you clarify what do you mean by “without diseases that could influence study findings and who could be measured”? Could you specify the exclusion criteria?

Results and discussion section:

  • In the discussion section you describe the influence of social norms and body composition in both sexes, greater social acceptance of large bodies in males and pressure on women to have a small body which increases the probability to an earlier identification of overweight in females (page 6 lines 213-216 and 239-244). It may be interesting to know if females have more prevalence of overweight/obesity than males in older ages (10-11). I have noticed that you describe the results of whole sample by age and by sex, but did you study the sample weight in relation to sex and age (prevalence of females with overweight/obesity in the older age group and prevalence of males with overweight/obesity in the same age group) ?

  • In Table 5, you describe the correlations between weight status, restriction, pressure to eat and monitoring. We can see how restriction and monitoring are positively related to children weight status (.231** and .125*, respectively) and pressure to eat is negatively related to the children weight (-.272**). However, in the same table pressure to eat is positively related to restriction (.176**). Moreover, in the discussion (page 7 lines 271-274) you affirm that “it has been detected that the pressure to eat healthy foods reduces children’s preference for them. Therefore, this also appears not to be a recommended practice.” (I understand that when children are under pressure, refuses healthy foods, they tend to eat junk food and increase weight).

So these results in relation to pressure to eat seem controversial. With the following sentence “although this could not be verified in our work” are you trying to clarify this controversy? Could you please clarify these results?

Author Response

REVIEWER 2

General comments: Thank you for the opportunity to review this manuscript. I find this study very interesting and I congratulate you for your work. It focuses on an important topic, relevant to public health and children health. Please see my comments below.

 Authors: Thank you very much for your comments. We honestly think that these comments have greatly improved the manuscript.

Introduction section:

1.- Page 1 line 30: According to the first reference cited, WHO reports an increase of overweight and obesity worldwide except in regions of Africa and Asia. So, I suggest to modify this sentence: “…Europe being the most affected region”.

 Response 1: Following the reviewer’s suggestion, we have changed the sentence, and we have changed the reference:

Lines 30-32

There are over 1.9 billion overweight/obese adults and over 380 million overweight/obese children in the world, with Europe being one of the most affected region[1].

Lines 348-350

  • World Health Organization. Obesity and overweight: key task https://www.who.int/news-room/fact-sheets/detail/obesity-and-overweight

2.- Page 1 line 40: Remove the question mark after “eating behaviour of children

 Response 2: Following the reviewer’s suggestion, we have removed this question mark:

Line 42

It has been found that some of parental feeding practices on the eating behaviour of children may be counterproductive.

Materials and Methods section:

1.- Page 2 lines 89-90: Could you clarify what do you mean by “without diseases that could influence study findings and who could be measured”? Could you specify the exclusion criteria?

 Response 1: This exclusion criterion includes children who cannot being measured or weighted because of a disease, like paraplegia or a sprain (we used a bioimpedance scale). We have changed the sentence:

Lines 91-94

The inclusion criteria were 4-5 and 10-11 years-old children, and children who could be measured and weighted. Two children could not be measured and weighted because of paraplegia and a sprain.

Results and discussion section:

1.- In the discussion section you describe the influence of social norms and body composition in both sexes, greater social acceptance of large bodies in males and pressure on women to have a small body which increases the probability to an earlier identification of overweight in females (page 6 lines 213-216 and 239-244). It may be interesting to know if females have more prevalence of overweight/obesity than males in older ages (10-11). I have noticed that you describe the results of whole sample by age and by sex, but did you study the sample weight in relation to sex and age (prevalence of females with overweight/obesity in the older age group and prevalence of males with overweight/obesity in the same age group)?

 Response 1: Following the reviewer’s suggestion, we have studied this relation and we have modified table 2:

Line 117

Age

Total

4-5 years

10-11 years

Total

(4-5 y)

Male

Female

Total

(10-11y)

Male

Female

Underweight/

normal weight

79.4%

93.7%

94.5%

92.7%

69.8%

68.7

70.2%

Overweight

15.2%

3.1%

4.1%

1.8%

23.3%

21.9%

25.5%

Obesity

5.4%

3.1%

1.4%

5.5%

6.9%

9.4%

4.3%

2.- In Table 5, you describe the correlations between weight status, restriction, pressure to eat and monitoring. We can see how restriction and monitoring are positively related to children weight status (.231** and .125*, respectively) and pressure to eat is negatively related to the children weight (-.272**). However, in the same table pressure to eat is positively related to restriction (.176**). Moreover, in the discussion (page 7 lines 271-274) you affirm that “it has been detected that the pressure to eat healthy foods reduces children’s preference for them. Therefore, this also appears not to be a recommended practice.” (I understand that when children are under pressure, refuses healthy foods, they tend to eat junk food and increase weight).

Response 2: The relation between pressure to eat and restriction is that there is a tendency to press to eat healthy food and restrict unhealthy, independently of weight status. But, our study and others show when a child has underweight, parents use more pressure to eat because they are concerned about the risk of malnutrition. However, parents use more restrictions to decrease food intake when their kid has overweight/obesity. It has been observed that pressure to eat causes children reject healthy eating because it is imposed. Thus, it would suggest that is not a good practice.

In order to clarify this idea, we added the following sentence:

Lines 284-285

Thus, it would be necessary to use a tool that allows to measure children’s eating preferences and their relationship with parent feeding practices.

3.- So these results in relation to pressure to eat seem controversial. With the following sentence “although this could not be verified in our work” are you trying to clarify this controversy? Could you please clarify these results?

Response 3: In our study, we studied the variable of pressure to eat, that analyses how parents press their children to eat in general, not to eat healthy food. Our results show a relation between underweight and pressure to eat, but they do not allow us to know if parents press to eat healthy or junk food. Thus, we are not able to verify if children refuse to eat healthy food because of pressure to eat this food.

Following the reviewer’s suggestion, we have added this sentence in the discussion section:

Lines 284-285

Thus, it would be necessary to use a tool that allows to measure children’s eating preferences and their relationship with parent feeding practices.

Reviewer 3 Report

Review of manuscript titled: “Parental perception of weight and feeding practices in schoolchildren: a cross-sectional study”

The first reference link is not working. The statement “There are over 1.9 billion overweight/obese adults and over 380 million overweight/obese children in the world, with Europe being the most affected region” needs a solid reference.

There are several “?” in the manuscript that are out of place.  The “have” in first sentence of the abstract should be “has”.  The manuscript needs a detailed check of the English language usage.  The following sentence does not make sense. “We invite to participate 622 children, 304 pupils aged 4 to 5 and 318 aged 10-11.”  Were 622 children invited and 304 children participated?

The self-selection bias of participants in the study, along with less than 50% participation calls into question the findings.  This is particularly true in 4-5 age group as there were only 8 overweight/obese participants. 

Although the study findings are of interest, the low number of overweight/obese participants’ calls into question the validity of the parental feeding findings. A larger study would validate the findings.  Parental perception of weight is interesting but it is well documented in the literature.

Author Response

REVIEWER 3

1.- The first reference link is not working. The statement “There are over 1.9 billion overweight/obese adults and over 380 million overweight/obese children in the world, with Europe being the most affected region” needs a solid reference.

 Response 1: Following the reviewer’s suggestion, we have changed the reference:

Lines 348-350

  • World Health Organization. Obesity and overweight: key task https://www.who.int/news-room/fact-sheets/detail/obesity-and-overweight

2.- There are several “?” in the manuscript that are out of place.

 Response 2: Thank you for your recommendation. We have removed this question mark

Line 42:

It has been found that some of parental feeding practices on the eating behaviour of children may be counterproductive.

3.- The “have” in first sentence of the abstract should be “has”.

Response 3: Thank you for reporting this mistake. Following the reviewer’s suggestion, we have changed this word:

Line 12

Childhood obesity has become a public health problem.

4.- The manuscript needs a detailed check of the English language usage.

Response 4: Thank you for your recommendation. A qualified native speaker for proper English language, grammar, punctuation and spelling has edited this manuscript, and the certificate has been submitted.

5.- The following sentence does not make sense. “We invite to participate 622 children, 304 pupils aged 4 to 5 and 318 aged 10-11.”  Were 622 children invited and 304 children participated?

Response 5: Following the reviewers suggestion, we have changed the sentence:

Lines 98-100

A total of 622 children were invited to participate, including 304 pupils aged 4 to 5 and 318 aged 10 to 11.

6.- The self-selection bias of participants in the study, along with less than 50% participation calls into question the findings.  This is particularly true in 4-5 age group as there were only 8 overweight/obese participants. 

Response 6:  In our study, the prevalence of overweight and obesity in 4- and 5-year-old children is lower than in the rest of the country, which may be due to the sample size and high non-response rate, especially in schools with low socioeconomic status. It is known that there is a relationship between the prevalence of overweight and obesity and socioeconomic status/level. Therefore, we have added these sentences in the discussion and limitations sections, and this reference in discussion section:

Lines 211-215:

The high rate of non- response among 4-5 year old pupils might have affected these results. It may also be that greater awareness of weight is affecting levels of overweight in younger children[36] or the non-response could be related with the fact that the schools with less participation were those which the socioeconomic level was low[38].

  1. Garrido-Miguel, M.; Cavero-Redondo, I.; Álvarez-Bueno, C.; Rodríguez-Artalejo, F.; Moreno, L.A.; Ruiz, J.R.; Ahrens, W.; Martínez-Vizcaíno, V. Prevalence and Trends of Overweight and Obesity in European Children from 1999 to 2016: A Systematic Review and Meta-analysis. JAMA Pediatr. 2019, 173, 1–13, doi:10.1001/jamapediatrics.2019.2430.

Lines 294-296:

Additionally, a large prevalence of parents with higher studies was found, due to the high rate of non-response among families of a low socio-economic level, which may have biased the findings along with the sample size.

7.- Although the study findings are of interest, the low number of overweight/obese participants’ calls into question the validity of the parental feeding findings. A larger study would validate the findings.  Parental perception of weight is interesting but it is well documented in the literature.

Response 7: Thank you very much for your comments. We honestly think that although the sample was small and biased, it could be the base for future studies with more representative samples in the province of Albacete and ensuring a representation of all socioeconomical status. In order to highlighting this, we have changed the following sentences in limitations and conclusion sections:

Lines 294-296:

Additionally, a large prevalence of parents with higher studies was found, due to the high rate of non-response among families of a low socio-economic level, which may have biased the findings along with the sample size.

Lines 325-329

More studies would be necessary to test these beliefs in Spain in the future, as social and cultural issues affect these relationships. Undertaking longitudinal studies to study causality would also be an interesting line of research with a representative sample size and ensuring a representation of families with all socioeconomic status.

Round 2

Reviewer 3 Report

This study adds to childhood obesity literature by providing a better understanding of the relationship between overweight/obesity in children and the practices and perceptions of parents.

However there are some items that need clarification. See below for details.

Line 92: Suggest changing from “To calculate the sample size, It was based on Comrey and Lee[24]…” TO “Sample size calculation was based on Comrey and Lee[24]…”

Lines 96 & 97: Suggest changing from “A total of 622 children were invited to participate, 304 pupils aged 4 to 5 and 318 aged 10 to 11.” TO “A total of 622 children, aged 4 to 5 and 318 aged 10 to 11, were invited to participate and 304 of those invited participated in the study.”

Tables 2 and 3 are grouped together.  They need to be separated.

Table 3 -  Is the relationship to greater prevalence of low or normal weight, significant between all 3 levels of education or just between certain levels?

Author Response

REVIEWER 3

This study adds to childhood obesity literature by providing a better understanding of the relationship between overweight/obesity in children and the practices and perceptions of parents.

However there are some items that need clarification. See below for details.

1.- Line 92: Suggest changing from “To calculate the sample size, It was based on Comrey and Lee[24]…” TO “Sample size calculation was based on Comrey and Lee[24]…”

 Response 1: Following the reviewer’s suggestion, we have changed the sentence:

Lines 93-95

Sample size calculation was based on Comrey and Lee[24] to the validation and adaptation of questionnaires, the main aim of the cross-sectional study.

2.- Lines 96 & 97: Suggest changing from “A total of 622 children were invited to participate, 304 pupils aged 4 to 5 and 318 aged 10 to 11.” TO “A total of 622 children, aged 4 to 5 and 318 aged 10 to 11, were invited to participate and 304 of those invited participated in the study.”

 Response 2: Following the reviewer’s suggestion, we have changed this sentence, but it is not correct. The correct data is:

Lines 97-99:

A total of 622 children, 304 aged 4 to 5 and 318 aged 10 to 11, were invited to participate and 316 of those invited participated in the study.

3.- Tables 2 and 3 are grouped together.  They need to be separated.

Response 3: Following the reviewer’s suggestion, we have moved table 2:

Lines 173-177

The prevalence of overweight and obesity was much higher among children of 10 and 11 years old (χ²= 30.147, p<0.00) (table 2).

Table 2. Weight status by age and sex.

Age

Total

4-5 years

10-11 years

Total

(4-5 y)

Male

Female

Total

(10-11y)

Male

Female

Underweight/

normal weight

79.4%

93.7%

94.5%

92.7%

69.8%

68.7%

70.2%

Overweight

15.2%

3.1%

4.1%

1.8%

23.3%

21.9%

25.5%

Obesity

5.4%

3.1%

1.4%

5.5%

6.9%

9.4%

4.3%

As shown in Table 3, a high level of education is related to greater prevalence of low or normal weight, both in fathers (χ²= 12.858, p=0.012) and mothers (χ²= 22.153, p=0.000).

Table 3. Weight status by parental educational level.

Father educational level

Mother educational level

Primary

Middle

University

p

Primary

Middle

University

p

Underweight/normal weight

33.2%

28.4%

38.4%

.012

19.0%

27.4%

53.6%

.000

Overweight

42.2%

33.3%

24.4%

29.8%

46.8%

23.4%

Obesity

76.9%

7.7%

15.4%

50.0%

18.8%

31.3%

4.- Table 3 -  Is the relationship to greater prevalence of low or normal weight, significant between all 3 levels of education or just between certain levels?

Response 4: Thank you for your comment. The relationship is significant between all weight status and all educational level, not only with low and normal weight. We have modified table 3 for its better understood:

Father educational level

Mother educational level

Primary

Middle

University

p

Primary

Middle

University

p

Underweight/normal weight

33.2%

28.4%

38.4%

.012

19.0%

27.4%

53.6%

.000

Overweight

42.2%

33.3%

24.4%

.012

29.8%

46.8%

23.4%

.000

Obesity

76.9%

7.7%

15.4%

.012

50.0%

18.8%

31.3%

.000
